# DOA Estimation in B5G/6G: Trends and Challenges

**DOI:** 10.3390/s22145125

**Published:** 2022-07-08

**Authors:** Ningjun Ruan, Han Wang, Fangqing Wen, Junpeng Shi

**Affiliations:** 1School of Electronic and Information, Yangtze University, Jingzhou 434023, China; ruannj@yangtzeu.edu.cn; 2College of Computer and Information Technology, China Three Gorges University, Yichang 443002, China; 3College of Physical Science and Engineering, Yichun University, Yichun 336000, China; hanwang@jxycu.edu.cn; 4Hubei Key Laboratory of Intelligent Vision Based Monitoring for Hydroelectric Engineering, China Three Gorges University, Yichang 443002, China; 5College of Electronic Countermeasure, National University of Defense Technology, Hefei 230037, China; shijunpeng20@nudt.edu.cn

**Keywords:** DOA estimation, massive MIMO, array signal processing

## Abstract

Direction-of-arrival (DOA) estimation is the preliminary stage of communication, localization, and sensing. Hence, it is a canonical task for next-generation wireless communications, namely beyond 5G (B5G) or 6G communication networks. Both massive multiple-input multiple-output (MIMO) and millimeter wave (mmW) bands are emerging technologies that can be implemented to increase the spectral efficiency of an area, and a number of expectations have been placed on them for future-generation wireless communications. Meanwhile, they also create new challenges for DOA estimation, for instance, through extremely large-scale array data, the coexistence of far-field and near-field sources, mutual coupling effects, and complicated spatial-temporal signal sampling. This article discusses various open issues related to DOA estimation for B5G/6G communication networks. Moreover, some insights on current advances, including arrays, models, sampling, and algorithms, are provided. Finally, directions for future work on the development of DOA estimation are addressed.

## 1. Introduction

Beamforming is one of the most attractive features of array signal processing. Thanks to beamforming techniques, one can enhance the desired spatial signal while restraining unwanted spatial interference and clutter. Before expected beams are designed, the direction-of-arrival (DOA) of the desired signal must be known in advance. Thus, DOA estimation is vital to array signal processing [1,2,3,4,5]. DOA estimation using sensor arrays refers to measuring the angles of the incoming source signals via colocated sensors. In general, the sensors are distributed in a given geometry. Equivalently, the incoming signals are measured in both the spatial domain and temporal domain. In fact, DOA estimation mainly depends on the geometry of the sensors. The path-differences between the sensors lead to various phase differences. The phase differences of all the sensors and the reference sensor consist of the spatial response vector. Empirically, the spatial response vector and the DOA are one-to-one in nonlinear mapping. Consequently, DOA estimation using spatial discrete measurements is a nonlinear problem. To tackle this issue, extensive efforts have been undertaken in past decades.

Next-generation wireless communications, called beyond 5G (B5G) or 6G, are envisioned to be multi-purpose systems that will be able to provide control, computing, communication, localization, and sensing (3CLS) services for the user [6]. The crucial techniques required to deliver 3CLS are DOA estimation and beamforming, which are known as intelligent physical layer techniques. To increase the spectral efficiency of the area, high expectations have been placed on both large-scale multiple-input multiple-output (MIMO) and millimeter wave (mmW) bands [7]. The former is characterized by massive colocated sensors, while the latter features bands that are located at extremely high frequencies (24 GHz–86 GHz). Usually, massive MIMO systems integrated with mmW frequency bands or sub-mmW frequency bands are known as mmW-MIMO systems. A simple beamforming example in a futuristic B5G/6G system is illustrated in Figure 1. Herein, only two applications are considered, namely, mobile users and unmanned vehicles. To provide low-latency and high-speed data services for the terminals under severe attenuation, the DOA of the terminals must be known to the base station in advance in order. Therefore, it is crucial for the base station to know the DOA of the terminals. 

In the literature, it has previously been reported that angle resolution is proportional to the sensor aperture. Meanwhile, in order not to introduce angle ambiguity, the inter-element distance d should be smaller (or equal to) than the half-wavelength. To support mmW communications in order to achieve faster, safer, and more reliable applications, large-scale antenna arrays with tens to hundreds of antennas must be adopted [8]. Benefiting from large-scale antenna arrays, mmW-MIMO systems are able to achieve tunable narrow beams as well as high gains, combating severe fading in the mmW bands. Consequently, mmW-MIMO systems can improve the resolution of DOA estimations. Unfortunately, DOA estimation is challenging in mmW-MIMO systems. On the one hand, the extremely small distance between the sensors of mmW-MIMO systems results in a small Rayleigh distance. As a result, DOA estimation algorithms may fail to work due to the coexistence of a near-field source and a far-field source. On the other hand, the mutual coupling effect of the mmW or sub-mmW caused by the half-wavelength spacing of the sensor array cannot be neglected, resulting in a mismatch between the ideal estimation model and the real one and yielding inaccurate DOA estimations or even invalid estimations. Furthermore, a massive sensor array means a huge data volume and a heavy computational burden for DOA estimation, which may not be implemented in real-time, especially in the case of two-dimensional (2D) DOA estimation. In other words, the hybrid-field sources, the mutual coupling effect, and the sparse sampling frameworks are the three challenges affecting DOA estimation in mmW-MIMO systems. Extensive efforts have been devoted to these three topics in the past decade.

In this paper, we will focus on hybrid-field sources, the mutual coupling effect, and sparse sampling frameworks in DOA estimation. More specifically, the contributions of this paper are as follows:(1)Data models representing DOA estimation with respect to hybrid-field sources, the mutual coupling effect, and sparse sampling frameworks have been introduced. The impacts of the corresponding model errors on DOA estimation have been analyzed.(2)The current advances in hybrid-field sources, the mutual coupling effect, and sparse sampling frameworks have been reviewed. Moreover, the limitations of state-of-the-art works have been summarized, and possible trends have been pointed out.

The rest of the paper is organized as follows: In Section 2, we present a signal model for DOA estimation. Section 3 reviews DOA estimation with hybrid-field sources. The mutual coupling issue is discussed in Section 4. The issue of DOA estimation with compressive sampling is discussed in Section 5. Finally, conclusions are provided in Section 6.

## 2. Signal Model of DOA Estimation

Consider a sensor array with *M*-element sensors. Suppose that there are *K* uncorrelated narrow-band sources impacting the far-field of the array (the correlated sources and wide-band sources are topics of interest [9] but are beyond the scope of this paper). Moreover, the signal and noise are uncorrelated, and both of them are modeled as independent complex Gaussian random processes. The array response of all of the sensors can be expressed as
(1)yt=∑k=1Kaθk,φk,Δkskt+Nt=Aθ,φ,ΔSt+Nt
where t denotes the snapshot index, aθk,φk,Δk∈ℂM×1 denotes the spatial response corresponding to the kth 1≤k≤K source signal, skt denotes the *k*th source signal, and θk and φk account for the elevation angle and azimuth angle of the kth source signal, respectively. Δk represents the range between the array and the kth source. Nt denotes the noise measurements. Aθ,φ,Δ=aθ1,φ1,Δ1,aθ2,φ2,Δ2,⋯,aθK,φK,ΔK∈ℂM×K denotes the spatial response matrix. In St=s1t,s2t,⋯,sKtT, the superscript *T* means transpose. Most current works focus on one-dimensional DOA estimation with far-field sources. Under such an assumption, the signal model in Equation (1) can be modified as
(2)yt=∑k=1Kaθkskt+Nt=AθSt+Nt
where aθk denotes DOA that is only dependent on the direction response vector, the m-th (m=1,2,⋯,M) entity of which represents the phase difference between the m-th sensor and the reference sensor. Aθ is DOA that is only dependent on the direction response matrix. It is well known that Aθ is related to the geometry of the sensor array. Moreover, in the presence of a specific sensor geometry, e.g., the uniform linear array (ULA), coprime array, or nested array, estimation problems are more easily solved. For example, the spatial response vector of the ULA can be approximated as
(3)aθk=1,e−j2πdsinθk/λ,⋯,e−j2πM−1dsinθk/λT
where d is the inter-element spacing between the sensors. Consequently, Aθ is a Vandermonde matrix, and the noiseless covariance matrix of yt is a Toeplitz matrix. 

DOA estimation is the preliminary purpose of beamforming and localization. DOA-based cooperative positioning applications been the subject of extensive interested in the past few years [10,11,12,13,14,15]. The cooperative positioning principle is depicted in Figure 2. To obtain the 2D position of the source $S_{0}$, at least two base stations, $B_{1}$ and $B_{2}$, are required. Moreover, the distance between the two base stations must be known in advance. Once the one-dimensional (1D) DOA has been achieved with respect to the two base stations ($\alpha_{1}$ and $\alpha_{2}$), the position of the source can be calculated via the location relationship between the source and the base stations. Similarly, the three-dimensional (3D) position of the source can be measured with two base stations. Herein, the coordinates of the two base stations must be known in advance. Moreover, the 2D-DOA of the source should be measured by the two base stations.

Super-resolution DOA estimation for far-field sources has a long history. Multiple signal classification (MUSIC) is recognized as pioneering the DOA algorithm [16]. To avoid an exhaustive search in MUSIC, the estimation of signal parameters via rotational invariance technique (ESPRIT) was developed [17]. This technique provides closed-form expression for DOA estimation. Improved representative algorithms include the root-MUSIC [18], beamspace-MUSIC [19], and weighted-MUSIC [20] algorithms; the propagator method (PM) [21]; and the matrix pencil [22] and sparsity-aware approaches [23,24]. Their advantages/disadvantages are shown in Table 1. However, it should be noted that most current works mainly focus on DOA estimation with an ideal background, e.g., a well-calibrated sensor array, far-field sources, and known source number. In practical applications, errors always exist. In the presence of errors, it is possible for model mismatches to occur between the ideal one and the true one, resulting in decreased estimation performance or even failed algorithms. In what follows, we will discuss three typical errors caused by hybrid-field, mutual coupling, and non-uniform sampling. 

## 3. Far-Field, Near-Field, and Hybrid-Field

In array signal processing, the Rayleigh distance is an important boundary that can be used to distinguish the near-field (or Fresnel zone) and far-field (or Fraunhofer region) sources. The Rayleigh distance *L* is defined as $L=2D^{2}/\lambda$, where *D* denotes the aperture of the sensor array, and λ accounts for the wavelength [25]. The source signal can be regarded as a far-field source if the distance between the array and the source is larger than *L*; otherwise, it can be treated as near-field source, as shown in Figure 3. Usually, the amplitudes of the sensor responses are related to the distances between the source and the sensors. In the presence of a far-field source, the wavefronts can be approximated as planar waves. However, in the presence of a near-field source, the wavefronts are spherical waves. 

To simplify DOA estimation models, there are some common assumptions that are usually made. It is usually assumed that the amplitudes of various sensors are equal to each other, and the phase-difference is approximated using their Taylor series. For far-field source scenarios, only first-order Taylor series are considered, in which the cosine direction is linear to differences in the array phase. Nevertheless, for near-field source situations, the second-order Taylor series are retained [26], and this is known as the Fresnel model. Herein, the difference in the array phase is a nonlinear function of the cosine direction and the distances between the source and the sensors. Compared to far-field source scenarios, an additional DOA-range item appears in the phase difference. An exact response vector is shown below [27].
(4)$$a^{\prime }\left(\theta_{k}\right)={\left[1,\left|r_{k}/r_{2,k}\right|\cdot e^{-j2\pi \left(r_{k}-r_{2,k}\right)/\lambda },\cdots ,\left|r_{k}/r_{M,k}\right|\cdot e^{-j2\pi \left(r_{k}-r_{M,k}\right)/\lambda }\right]}^{T}$$
where $r_{k}$ denotes the distance between the kth source signal and the reference sensor, and $r_{m,k}$ denotes the range between the kth source signal and the mth sensor.

By treating near-field source estimation issues as a joint DOA–range estimation problem, the traditional MUSIC algorithm can be migrated directly [28] and which will suffer from high-dimensional grid searching. To reduce the computational burden, a higher order cumulant is often adopted to construct a new rotational invariance relationship [29] that only dependent on the DOAs of the sources. In addition, electromagnetic vector sensors (EMVSs) are a promising alternative [30,31,32] since they can avoid the spherical wavefronts of the near-field sources via the temporally rotational invariance characteristic of EMVS measurements. Unlike traditional scalar sensors, a complete EMVS consists of six mutual orthogonal components, three of which are electronic field responses, and three of which are magnetic field responses. Each component of the EMVS is a function of the elevation angle θ, azimuth angle φ, auxiliary polarization angle γ, and polarization phase difference η, i.e.,
(5)$$\begin{array}{ll} b\left(\theta ,\varphi ,\gamma ,\eta \right) & ={\left[E_{x},E_{y},E_{z},H_{x},H_{y},H_{z}\right]}^{T}\\ & =\left[\begin{array}{cc} -\sin\varphi & \cos\theta \cos\varphi \\ \cos\varphi & \cos\theta \sin\varphi \\ 0 & -\sin\theta \\ \cos\theta \cos\varphi & \sin\varphi \\ \cos\theta \sin\varphi & -\cos\varphi \\ -\sin\theta & 0 \end{array}\right]\left[\begin{array}{c} \cos\gamma \\ \sin\gamma e^{j\eta } \end{array}\right] \end{array}$$

The situation becomes more complicated in the presence of hybrid-field sources. A common pre-processing method regards far-field sources as ‘+∞‘ near-field sources [25]. The estimation algorithms for hybrid-field sources can be divided into two categories according to the estimation strategies being implemented: separated estimation methods and unified estimation approaches. The former estimates the parameters of the far-field sources (or the near-field sources) and then mathematically removes the components associated with the far-field sources (or the near-field sources) from the array measurements (e.g., oblique projection, reconstruction, differencing). Thereafter, it obtains the parameters corresponding to the near-field sources (or far-field sources). The latter first achieves the DOAs of all of the sources. With respect to each of the previously obtained DOAs, it then estimates the corresponding range between the source and the sensor. Finally, it partitions the sources into near-field sources and far-field sources according to the estimated ranges.

It should be pointed out that the currently existing DOA estimation frameworks for hybrid-field sources are based on approximated data models, which create mismatching in the real one and yield decreased estimation performance. To deal with hybrid-field sources, more exact models are necessary. Nevertheless, only a few efforts have been made to conduct DOA estimation with an exact model. In [33], the author provides an exact data model for near-field sources, revealing the relationship between the amplitude/phase and the DOA range of the sensors. This indicates that hybrid-field sources can be separated first and undergo parameter estimation later. From our point of view, hybrid-field sources will be an unavoidable issue in B5G/6G, and exact model-based DOA estimation frameworks will receive continuous attention. To deal with hybrid-sources in DOA estimation, the following concerns should be stressed:

(1)Accurate estimations of the number of sources with respect to far-field sources and near-field sources. Most of the current algorithms rely on prior knowledge of the source number. However, in the presence of hybrid-field sources, the existing algorithms can only estimate the total number of hybrid-field sources, while current algorithms require exact numbers with respect to far-field sources and near-field sources. An inaccurate source number would lead to model mismatching between the ideal model and the actual model and would yield decreased performance.(2)The near-field sources should be carefully separated from the far-field sources. After the source number has been determined, the near-field signals need to be separated from the far-field signals correctly; otherwise, the estimation algorithms may fail to work.(3)Super-resolution parameter estimation based on the exact model. Current DOA estimation algorithms for hybrid-field sources rely on the approximation of the Fresnel model, which has been proven to be unreasonable [34]. For accurate DOA estimation, extensive efforts need to be devoted to data modeling, array design, and estimation algorithms.

## 4. Sensor Mutual Coupling

The mutual coupling effect is a common model error in collocated sensors [35]. Essentially, mutual coupling is caused by the radiation effect of the sensors in the stimulated electromagnetic signal received by other sensors. For an M-element sensor array, the mutual coupling effect between sensors can be described by an M×M mutual coupling matrix C, the (m,n)th entity of which reveals the mutual coupling coefficient between the mth sensor and the nth sensor. The previous literature indicates that the mutual coupling strength between the two sensors is inversely proportional to the distance between them, and the mutual coupling coefficient of two sensors that are far apart can be approximated as zero [36]. Put simply, the mutual coupling matrix is a symmetric matrix. 

Specially, the mutual coupling matrix corresponding to a traditional ULA can be modeled as a banded symmetric Toeplitz matrix since the mutual coefficients between sensors with the same range are the same, as shown in Figure 4a. Consider a special case in which the mutual coupling effect only exists between the two adjacent sensors and the mutual coupling coefficients only depend on the distance between the sensors. Then, the mutual matrix can be approximated by
(6)$$C=\left[\begin{array}{ccccccc} 1 & c_{1} & 0 & 0 & 0 & 0 & 0\\ c_{1} & 1 & c_{1} & 0 & 0 & 0 & 0\\ 0 & c_{1} & 1 & c_{1} & 0 & 0 & 0\\ 0 & \ddots & \ddots & \ddots & \ddots & \ddots & 0\\ 0 & 0 & 0 & c_{1} & 1 & c_{1} & 0\\ 0 & 0 & 0 & 0 & c_{1} & 1 & c_{1}\\ 0 & 0 & 0 & 0 & 0 & c_{1} & 1 \end{array}\right]$$

In practical applications, the mutual coupling strength may be related to the DOA of the incoming sources [37,38,39], and mutual coupling will be more complicated. In the presence of coupling errors, the signal model in Equation (2) can be updated to be
(7)$$y\left(t\right)=\sum_{k=1}^{K}C_{k}a\left(\theta_{k}\right)s_{k}\left(t\right)+N\left(t\right)$$
where $C_{k}$ denotes the mutual coupling matrix corresponding to the kth source. Namely, model mismatch would occur in DOA estimation, thus leading to decreased estimation performance. Typical mutual coupling calibration approaches include electromagnetic calculation methods (such as the moment method [40] and the uniform theory of diffraction [41]), active calibration methods (auxiliary source signal and off-line computing [42]), and autocalibration methods (on-line computing). Among the approaches listed above, the self-calibration method is the most attractive because it does not have additional hardware costs. Up until now, extensive efforts have been devoted to the auto calibration approaches for mutual coupling. For examples, see [43,44,45] and the references therein. However, most of the methods focus on the mutual coupling issue of linear arrays, and a few works are concerned with the mutual coupling issue in nonlinear arrays [46,47], which is more complicated. However, since the mutual coupling coefficients in practical applications are sensitive to the humidity, temperature, and even the DOA of the incoming source, the above methods are hardly able to accurately estimate the mutual coupling matrix.

As pointed out earlier, the mutual coupling between two sensors that are far apart from each other can be ignored. Therefore, a straightforward strategy to avoid mutual coupling is the use of a spare array, i.e., the inter-element spacing between sensors is much larger than a half wavelength. However, sparse array geometry would bring ambiguity to DOA estimation, yielding invalid estimates. Two methodologies are capable of eliminating such ambiguity. One is the specific array manifold method, such as the coprime array [48]. A coprime array consists of an M-element ULA and an N-element ULA, where M and N are coprime integers, and the two ULAs share a sensor, as depicted in Figure 4b. The inter-element distances of the two ULAs are N and M, respectively. Although the DOAs obtained from the two sparse ULAs are ambiguous, the unambiguous DOA can be uniquely determined via the coprime characteristic between the two subarrays. The above spare geometry can be easily extended to a 2D case [49]. Since sparse arrays occupy less sensors than traditional uniform arrays, they can identify less sources than uniform arrays. Moreover, sensors in a sparse architecture must be strictly constrained to known positions, otherwise complicated calibration must be carried out.

Another alternative is the EMVS array, which is depicted in Figure 4c. As we emphasized earlier, traditional scalar sensors can only sense the profiles of electromagnetic waves, while a complete EMVS can measure all of the electromagnetic profiles of a signal. The sparse EMVS array geometry can avoid the mutual coupling effect, and it can provide unambiguous 2D-DOA estimations by combining the uniqueness of the DOA estimations with a single EMVS. Moreover, DOA estimation using the complete collocated EMVS array is insensitive to the sensor positions, making it possible to be conformal with the platform. The advantages and disadvantages of the sparse scalar/vector sensor array are listed in Table 2.

Since both the sparse array and EMVS array are two developing solutions, there is plenty of room for further research. From the viewpoint of the authors, the following aspects may be particularly interesting to the readers the present work: (1)For sparse array geometry, how to reduce the redundancy of sensors (or how to maximize the aperture with a given number of sensors) and how to unify (generalize) the array design requires sustained attention. Regardless of whether coprime arrays or nested arrays are used, two-level sparse arrays exhibit a common feature, i.e., the inter-element spacings of two subarrays are coprime. In other words, the important array parameters are fixed once the array type has been chosen. There is a need to develop more a general formulation for previous array parameters. Moreover, more efficient super-resolution algorithms for sparse array geometry is another topic that requires further research.(2)For the EMVS array, optimization is necessary for a less redundant EMVS [50]. As we pointed out in Equation (5), a single EMVS consists of six components, but only four unknown parameters are involved. Therefore, the EMVS array is redundant. The design of a less redundant EMVS and EMVS array are two interesting topics. Additionally, how to estimate high-resolution 2D-DOAs combined with the array geometry is also a topic of interest.

## 5. Spatial/Temporal Compressive Sampling

Signal sampling is a fundamental task in DOA estimation. Traditional signal acquisition has been dominated by the Shannon–Nyquist principle for more than seventy years. According to Shannon–Nyquist principle, to recover a narrow-band signal with cut-off frequency $f_{c}$, the sampling frequency should be at least twice the value of $f_{c}$. The massive sensor arrays in B5G/6G wireless communications imply the urgent need for a multi-channel analog-to-digital converter (ADC). This also introduces new challenges to data transmission, storage, and processing. Compared to complicated single sampling and massive volume data, the DOA of the sources is usually sparse in the spatial domain, which implies that there is an imbalance between DOA estimation and traditional sampling architecture. To break such imbalance, the concept of compressive sampling (CS) was introduced. According to CS, a sparse single or an approximately sparse signa it can be reconstructed from its sub-Nyquist measurement with a high degree of probability [51]. A CS-based framework can reduce the complexity of the hardware required for DOA estimation, and it is one of the most attractive research areas in B5G/6G wireless communications. Since DOA estimation mainly relies on spatial-temporal measurements, CS-based DOA estimation frameworks can be divided into three categories, namely spatial CS frameworks [48,52,53,54,55,56,57,58,59], temporal CS frameworks [60,61,62,63], and spatial-temporal CS frameworks [64]. 

The purpose of a spatial CS framework is to reduce the number of the front-end circuit chains. Mathematically, a spatial CS-based DOA framework for an N-element sensor array can be formulated as
(8)xt=Φyt=ΦAθSt+Et
where Φ∈ℂM×N is the reduced-dimensional transform matrix, and yt denotes the original array signal in Equation (2). A general framework for a spatial CS framework is provided in Figure 5. Herein, a linear combining network consists of several summation units, and the number of the summation units is much smaller than the number of sensors. In each summation unit, signals with different weights are added with respect to various sensors. Thereafter, the compressed signals are sampled via ADCs, after which, DOA estimation can be accomplished via solving a sparse inverse problem.

In a spatial CS framework, the linear combining network plays an important role in compressing spatial array measurements. The combining network is the selective matrix, which forms a sparse array geometry (as seen in coprime arrays [48] and random arrays [52]) using the traditional ULA. Usually, combining networks can be divided into three categories according to the combination type, namely the fully connected architectures, the sub-connected architectures, and the switch-based architectures. In the fully connected structures, each RF chain is connected to all of the antenna elements through phase shifters. In this type of architecture, how to design the combining network is an important issue. In [53], the compressive matrix is randomly selected according to the Gaussian distribution. In [54], the compressive matrix is optimized based on a prior probability distribution. In [55], the author analyzes the tight sufficient conditions for the compressive matrix. In sub-connected architectures, all of the receiving antennas are divided into several groups, and each RF chain is only connected to a subset of antennas. Therein, the subset mode is the kernel [56,57]. Switch-based architectures are highly analogous to fully connected structures, but the switches replace the phase shifters in the analog processing stages. In [58], a regularized complementary antenna-switching framework is introduced. In [59], the three architectures are combined into a unified expression in which the compressive matrix is regarded as a time-varying form. A dynamic maximum likelihood estimator was developed for DOA estimation.

In temporal CS frameworks, compressing occurs in the temporal domain, and the spatial structures remain. The purpose of a temporal CS is to reduce the sampling rate of the ADC or to reduce the bit rate of the measurement. An example of a temporal CS is presented in Figure 6, in which each channel signal is quantified with a one-bit ADC, i.e., the output of the ADC is 1 or −1. For example, for the n-th sensor, the one-bit quantization can be expressed as
(9)$$x_{m}\left(t\right)=O\left(y_{m}\left(t\right)\right)$$
where $O\left(\cdot \right)$ denotes the quantization processing of the complex value, which is given by
(10)$$O\left(z\right)=\frac{1}{\sqrt{2}}\left(sign\left(real\left(z\right)\right)+jsign\left(imag\left(z\right)\right)\right)$$
where if z > 0, $sign\left(z\right)$ returns 1; otherwise, it returns −1. $real\left(z\right)$ and $imag\left(z\right)$ return the real part and the imaginary part of the complex value, respectively. Consequently, the one-bit quantization results can be expressed as
(11)$$X\left(t\right)=O\left(AS\left(t\right)+N\left(t\right)\right)$$

Thereafter, DOA estimation can be achieved via solving an optimization issue using the low-resolution one-bit measurement. 

With the exception of one-bit quantification, various quantification networks have been discussed. In [61], the DOA estimation issue is considered under the conditions of a low-bit ADC, and a deep-learning network is implemented to eliminate quantization noise. In [62], a linear additive quantization noise model is considered for low-bit sampling DOA estimation issues, and an improved MUSIC method is introduced. The impact of low-resolution ADC on DOA estimation is analyzed in detail. In [63], a mixed-ADC structure is proposed. Therein, the sensor array is divided into two subsets, and the high-resolution ADC and the low-resolution ADC are separately applied in those subsets. A subspace-based estimator is utilized for DOA estimation.

In a spatial-temporal CS framework, the array signal is compressed in both the spatial and temporal domains simultaneously. An example of a spatial-temporal CS model is shown in Figure 7: the modular hybrid sampling method introduces extensive missing entities to the measurement matrix. The recovery of the complete measurement matrix can be interpreted as a matrix completion issue. Taking the noiseless array measurement X as an example, we can assume that the sampled measurement matrix is $\tilde{X}$, and the above recovery process can be formulated as
(12)$$\begin{array}{ccc} \min & & \mathrm{rank}\{X\}\\ s.t. & & S_{\Omega }(X)=\tilde{X} \end{array}$$
where $S_{\Omega }(X)$ denotes the down-sampling operation, which returns the data matrix with missing entities, after which the DOA can be estimated with the existing algorithms.

In [65], the hybrid CS architecture is extended to a massive MIMO system for 2D localization, and a matrix factorization variant is devised to recover the data matrix. Another hybrid CS architecture is proposed in [66], in which a combination of a one-bit ADC and sparse linear arrays is provided. 

It is worth noting that the hardware costs of CS-based frameworks are reduced at the expense of a more sophisticated computational burden or decreased DOA estimation performance. Consequently, the following areas are of interest for future research:

(1)The implementation of low-complexity hardware for CS-based architectures for DOA estimation. With the advent of cloud platforms for B5G/6G wireless communications, the calculated overhead can be ignored, indicating that that are bright prospects for CS in the near future. However, CS-based frameworks can be further improved in terms of the total hardware costs (power, volume, electromagnetic compatibility, etc.), the overall system clock frequency, and the number of ADC channels/rates. In addition, how to make CS-based architectures compatible with the modular sampling in the communication system is also of interest.(2)Efficient DOA estimation algorithms based on sub-Nyquist measurements are of interest for future interest. Usually, optimization-based DOA estimators are inefficient. Since B5G/6G enforce strict latency constraints, how to achieve real-time DOA estimation is of interest. Moreover, how to overcome issues related to the low signal-to-noise ratio (SNR) and how to deal with the coherent source issue are also relevant.

## 6. Conclusions

In this article, we introduced DOA estimation, one of the most attractive intelligent physical layer techniques, into 6G wireless communication networks. Three challenges have been rigorously discussed, namely the coexistence of far-field sources and near-field sources, the mutual coupling effect, and spatial/temporal CS. Possible solutions have been listed, and some promising research areas have been highlighted. In conclusion, DOA estimation is an attractive research topic, but there are still many issues that remain open for further investigation.

## Figures and Tables

**Figure 1 sensors-22-05125-f001:**
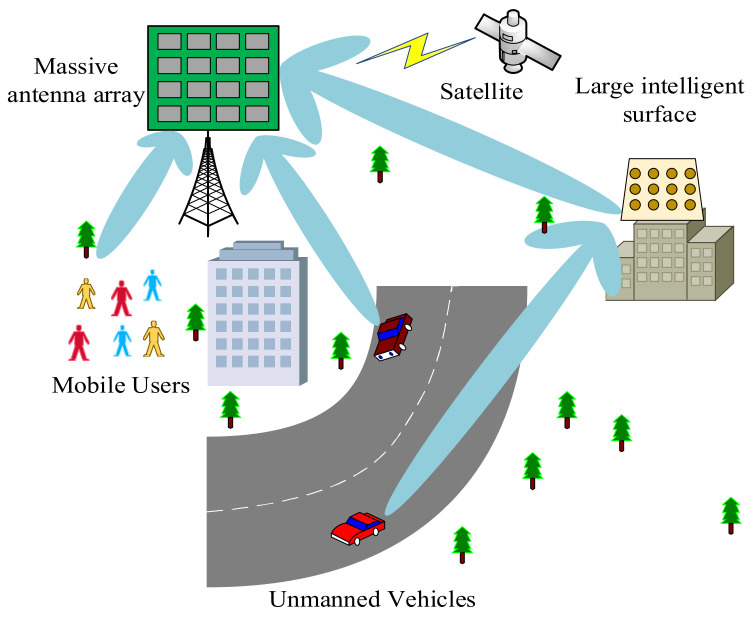
Illustration of beamforming in B5G/6G wireless communication networks.

**Figure 2 sensors-22-05125-f002:**
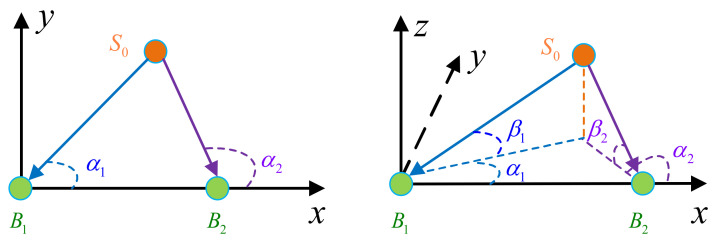
Schematic diagram of DOA-based positioning using two cooperative deciphers: (**left**) 2D positioning using 1D-DOA estimation; (**right**) 3D positioning using 2D-DOA estimation.

**Figure 3 sensors-22-05125-f003:**
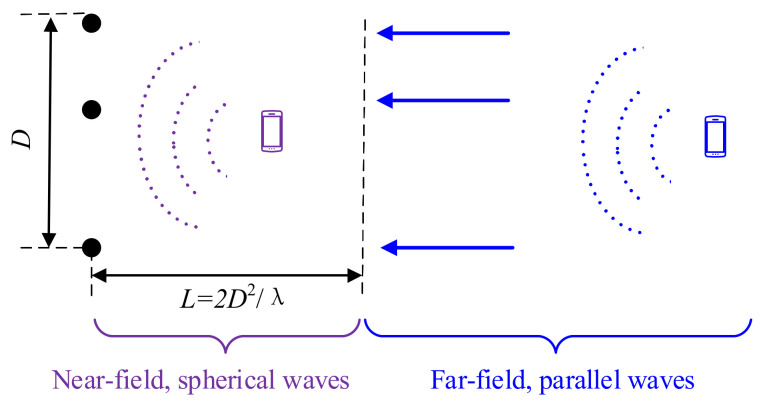
Illustration of near-field source and far-field source.

**Figure 4 sensors-22-05125-f004:**
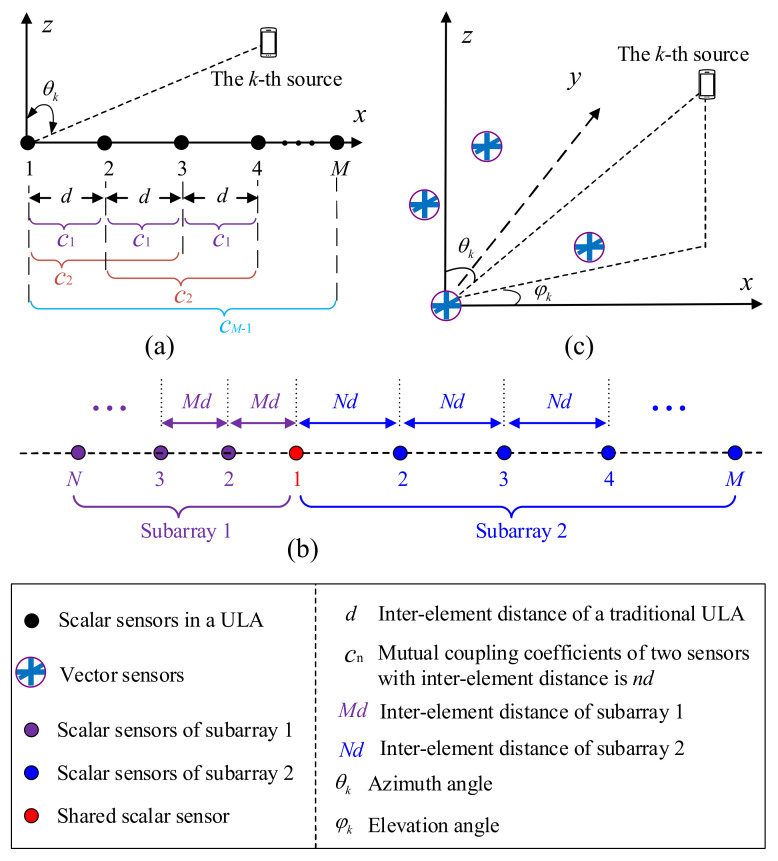
(**a**) Mutual coupling effect in DOA estimation with a ULA; (**b**) illustration of a coprime array; (**c**) EMVS array.

**Figure 5 sensors-22-05125-f005:**
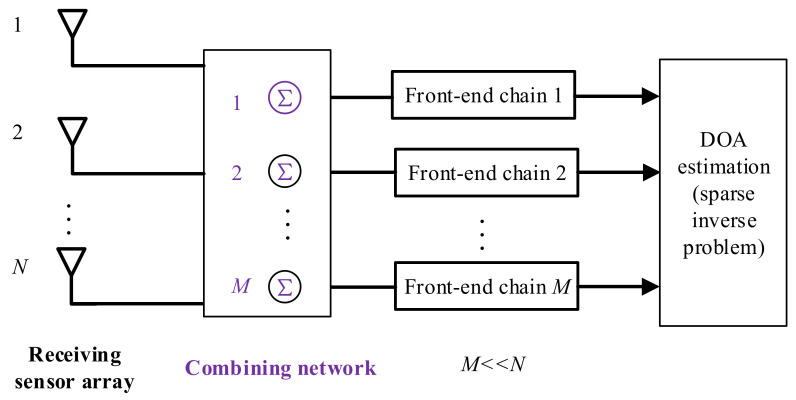
A general framework of a spatial CS for DOA estimation.

**Figure 6 sensors-22-05125-f006:**
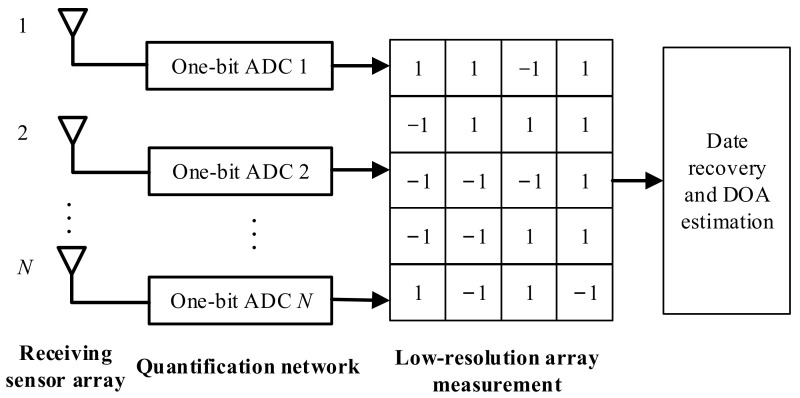
An example of a temporal CS for DOA estimation.

**Figure 7 sensors-22-05125-f007:**
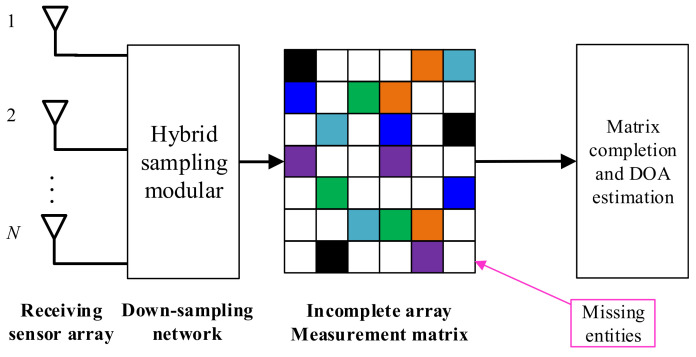
An example of a hybrid CS for DOA estimation.

**Table 1 sensors-22-05125-t001:** Advantages and disadvantages of typical algorithms.

Method	Advantages	Disadvantages
MUSIC	Suitable for arbitrary geometry	Computationally inefficient, off-grid issues (can be avoided by root-MUSIC)
ESPRIT	Closed-form solution	Only suitable for uniform array geometry
PM	Computationally economic	Sensitive to small snapshots
Matrix pencil	Suitable for single snapshot	Sensitive to noise power
Sparse algorithms	Super-resolution, insensitive to prior knowledge of source number	Computationally inefficient, off-grid issues

**Table 2 sensors-22-05125-t002:** Advantages and disadvantages of sparse scalar/vector sensor array.

Array Type	Advantages	Disadvantages
Scalar array	Low redundancy and low complexity	Strict sensor position, low identifiability
EMVS array	Closed-form solution, capable of 2D-DOA estimation with 1D geometry, suitable for arbitrary geometry, robust to sensor position error, better identifiability	High redundancy and high complexity

## Data Availability

Not applicable.

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
