# Peer review of "DOA Estimation in B5G/6G: Trends and Challenges"

_sensors, 2022, doi:10.3390/s22145125_

Round 1

Reviewer 1 Report

The authors have addressed all my concerns raised in the last round review. The presentation quality is improved. I am happy to accept this article.  

Author Response

Thanks for your encouragement.

Reviewer 2 Report

Thank the authors for addressing my previous comments. I am satisfied with the revisions. 

Author Response

Thanks for your comments in improving this paper.

Reviewer 3 Report

This is a good review for the DoA. However, it may be better to add other methods for future generation with other technology assist DoA, such as GPS, Beidou, etc. Highly integrated hybrid system should be possible in the future. 

Conclusion paragraph is in a way too concise. A summary of primary content is expected with informative description.

Author Response

Comment 1. This is a good review for the DoA. However, it may be better to add other methods for future generation with other technology assist DoA, such as GPS, Beidou, etc. Highly integrated hybrid system should be possible in the future. 

Reply: This is a very ineresting topic. Actually, DOA estimation has been widely applied in radars, wirelsess communications,  and also GPS. Since we mainly focus on the DOA estimation technology instead of its applications, those methods have not been included in the final revision.

Comment 2. Conclusion paragraph is in a way too concise. A summary of primary content is expected with informative description.

Reply: Thanks for your kindly reminder. In this paper, we have stressed three of the main issues in DOA estimation of massive MIMO systems. Some of the current solutions have been listed, and the advantagies/disvantagies have been analized. Since we have streed so many aspects and the conclusion senction is short, those details have been ignored in the conclusion section.

This manuscript is a resubmission of an earlier submission. The following is a list of the peer review reports and author responses from that submission.

Round 1

Reviewer 1 Report

This work reviews the determination of DOA, but does not combine the idea of model error. Including past research results and the latest research progress. Does not meet the requirements for a literature review. The most important point is that since it is a literature review, it needs a large number of references to support, and the number of references cited in this research work is too small. For the literature review problem, it is necessary to point out the challenges existing in the current research and the corresponding solutions. Moreover, abbreviations, such as DOA, MIMO, and THz, require full names when they first appear.

Reviewer 2 Report

   DOA (Direction Of Arrival) estimation, also known as spectral estimation (spectral estimation), angle of arrival (Angle Of Arrival) estimation. There are many possible sources a propagation path and the angle of arrival. If several transmitters simultaneously, a potential source for each multipath component at the receiver. Therefore, the receiving antenna can estimate the angle of arrival becomes very important, which aims to decipher the location of the transmitter at work as well as possible in which the transmitter. By measuring the radiation signal DOA (DOA stands for Direction Of Arrival) or the angle of arrival (AOA) to estimate the location of the radiation source. In this manuscript, the authors discussed various open issues on DOA estimation for next generation wireless communications. Moreover, they provided some insights on current advances, including array, model, sampling and algorithm. However, although the authors provide many theorical insights for the DOA estimation, but 

 the results are unclearly presented. For example, by using the DOA estimation theory presented by the authors, how does the receiving antenna estimate the angle of arrival, how dose decipher the location of the transmitter. Moreover, there are many knocking errors in the lines 235,74 and 70. Therefore, the manuscript is not suggested to be published in current forms.

Author Response

(Note from the assistant editor:  Dear Reviewer, you are marked as Reviewer 2 in the system. Thank you for your efforts.)

Reviewer 3 Report

The paper addresses three typical issues about DOA estimation in the mmW-massive-MIMO system, hybrid-field sources, mutual coupling, and spatial/temporal compressive sampling (CS), respectively. Some possible solutions have been listed and some promising research areas have been highlighted.

  1. The title of this paper is too general and not appropriate. This paper mainly discusses three issues of narrow-band DOA estimation in the mmW-MIMO system: hybrid-field sources, mutual coupling, and huge data volume burden, respectively. Simply summing up these three issues as "model errors" is inappropriate.
  2. In the sensor mutual coupling part, the typical mutual coupling calibration approaches are listed in lines 168-170 without citations, which should be added. Besides, citations [24-26] address DOA estimation with direction-dependent mutual coupling, so the claim in lines 170-172 is not appropriate.
  3. Only three outdated papers are cited in the spatial/temporal CS part, the newest advances and insights on this issue should be included.
  4. Solid papers should be cited and reviewedto back up the claim in line 49 that mmW-MIMO can improve the resolution of DOA estimation.
  5. The "small Rayleigh distance" should be changed into "large Rayleigh distance" in line 50. The phases in equation (4) are inaccurate and should multiply by 2.
  6. Theclaim that THz is featured by mmW or sub-mmW frequency band in line 41 is not accurate by definition.
  7. Figure 1 is not clearly drawn and well explained.
  8. The title of figure 2 and Figure 4-c are not right.
  9. A lot of grammar mistakes exist and should be thoroughly revised. For example, "yields" in line 54, "source" in line 60, "that caused" in line 96, "they" in line 264, contents in lines 149 and 256, and so on.

Reviewer 4 Report

The article provides an overview of DoA estimation with model error. Overall, it is well written. Some minor comments are given below 

-- The theme of the abstract and introduction do not seem to match the title. I suggest that the authors highlight the impact of model errors on DoA estimation in beyond 5G and 6G communications, and also the critical importance of solving the issue. 

The title can be improved; authors may consider something like:

DOA estimation with Model Errors in Beyond 5G and 6G: An Overview

-- "DOA estimation would suffer from mmW-MIMO system"

authors may consider saying "DOA estimation is challenging in mmWave-MIMO systems..."

-- (1): bold letter S(t) is not defined 

-- The square, which may be caused by font issue, often appears in equations throughout the article;  

-- Some recent works highlighting the importance of DOA estimation in mmWave massive MIMO communications are missing in the Introduction, such as 

"Recent Breakthroughs on Angle-of-Arrival Estimation for Millimeter-Wave High-Speed Railway Communication," in IEEE Communications Magazine, vol. 57, no. 9, pp. 57-63, September 2019

Round 2

Reviewer 2 Report

None

Reviewer 3 Report

In the revised manuscript, new references on antenna mutual coupling and compressed sampling are added. Title, definition, some grammar and obvious errors are corrected.

However, the following issues still exist:

1. The novelty and timeliness of the content are not strong. Many issues (e.g., mixed fields, compressed sampling) have been extensively studied in the literature, and the trends and challenges highlighted in this paper lack depth and novelty.

2. The paper lacks elaboration, comparative analysis and summary of the advantages and disadvantages of existing literature.

3. There are still some basic grammar and expression errors.